# Longitudinal analysis of *Plasmodium* sporozoite motility in the dermis reveals component of blood vessel recognition

Christine S Hopp[1]*, Kevin Chiou[2], Daniel RT Ragheb[1], Ahmed M Salman[3], Shahid M Khan[3], Andrea J Liu[2], Photini Sinnis[1]*

[1]Department of Molecular Microbiology and Immunology, Johns Hopkins Bloomberg School of Public Health, Baltimore, United States; [2]Department of Physics and Astronomy, University of Pennsylvania, Philadelphia, United States; [3]Department of Parasitology, Leiden Malaria Research Group, Leiden University Medical Center, Leiden, Netherlands

**Abstract** Malaria infection starts with injection of *Plasmodium* sporozoites by an *Anopheles* mosquito into the skin of the mammalian host. How sporozoites locate and enter a blood vessel is a critical, but poorly understood process. In this study, we examine sporozoite motility and their interaction with dermal blood vessels, using intravital microscopy in mice. Our data suggest that sporozoites exhibit two types of motility: in regions far from blood vessels, they exhibit 'avascular motility', defined by high speed and less confinement, while in the vicinity of blood vessels their motility is more constrained. We find that curvature of sporozoite tracks engaging with vasculature optimizes contact with dermal capillaries. Imaging of sporozoites with mutations in key adhesive proteins highlight the importance of the sporozoite's gliding speed and its ability to modulate adhesive properties for successful exit from the inoculation site.

*For correspondence: chopp1@ jhu.edu (CSH); psinnis@jhsph. edu (PS)

Competing interests: The authors declare that no competing interests exist.

## Introduction

Through the bite of an infected mosquito, the mammalian host is infected with *Plasmodium* sporozoites, which migrate through the skin to invade blood vessels. Sporozoites are then carried by the blood flow to the liver, where they initiate a liver- and subsequently a blood-stage infection (*Sinnis and Zavala, 2012*). Sporozoite motility, a substrate-dependent gliding motility, is essential for the exit from the dermis, and as a result for sporozoite infectivity (*Vanderberg and Frevert, 2004*; *Amino et al., 2006*; *Hellmann et al., 2011*; *Ejigiri et al., 2012*). In comparison to fast migrating mammalian cells, such as lymphocytes, which crawl at approximately 0.1 µm/s, sporozoites move at 1–3 µm/s (*Amino et al., 2006*; *Hellmann et al., 2011*; *Ejigiri et al., 2012*). Given the energy cost of such a remarkable speed, fast migration is likely key for sporozoite infectivity. Gliding speed is affected by the turnover of focal adhesion sites (*Münter et al., 2009*), as well as by environmental obstacles present in the dermis (*Hellmann et al., 2011*). While on two-dimensional substrates in vitro, salivary gland sporozoites glide in a circular pattern. In the dermis, this motion is transformed to a complex non-linear path (*Amino et al., 2006*; *Hellmann et al., 2011*).

Recent work has shown that both the circumsporozoite protein (CSP) and the thrombospondin-related anonymous protein (TRAP) have important roles for the exit of *Plasmodium berghei* sporozoites from the dermal inoculation site (*Coppi et al., 2011*; *Ejigiri et al., 2012*). Proteolytic processing of CSP leads to removal of the N-terminus and exposure of a cell-adhesion domain (*Coppi et al., 2011*). Sporozoites expressing a mutant CSP which lacks the N-terminus (CSΔN), thus mimicking the proteolytically processed form of CSP, display normal infectivity when inoculated intravenously (*Coppi et al., 2011*). However, when CSΔN sporozoites are injected intradermally, parasites are undetectable

**eLife digest** Malaria remains a devastating disease in many parts of the world. Malaria parasites enter the host via the skin, where they are deposited by infected mosquitoes as they look for blood. The parasites must exit the skin to reach the liver, where they multiply and ultimately infect red blood cells, where they cause the symptoms of the disease. In the skin, the parasites must move to find blood vessels that they enter to travel via the blood circulation to the liver. Only about 10–20% of parasites make it out of the skin, making this a bottleneck for the parasite.

Scientists have been working to develop vaccines that would protect people against malaria. One way these could work would be to stop malaria parasites from leaving the skin and entering the blood vessels. But to do that, more needs to be learnt about how the parasites move in the skin and enter the blood vessels.

Hopp et al., using a mouse model of malaria, created malaria parasites that produce a fluorescent protein that allows the parasites to be tracked after they have been injected into the skin of a mouse's ear. This revealed that the parasites have two ways of moving. After first being injected, the parasites move quickly and freely. The parasites slow down when they come close to a blood vessel and move on or around the vessel for some time before entering it. During this stage of movement, the parasites tend to move in paths that follow the curvature of the blood vessels, which may improve how well they make contact with the blood vessel surface and may enable them to find the areas of the vessels best suited for entry.

Next, Hopp et al. investigated how two parasite mutants move through mouse skin. Both mutants had previously been found to be less likely than wild-type parasites to exit the inoculation site. Hopp et al. found that one of the mutants moves slowly after being injected and so explores a smaller tissue volume than normal and encounters fewer blood vessels. The second mutant parasite spends more time than normal moving on the surface of the blood vessels, but finds it difficult to enter them.

Continuing this work will allow us to learn more about the interactions between the parasite and the blood vessels, which in turn could reveal key events that could be targeted by a vaccine. Furthermore, the significant amount of time that the parasites spend moving and looking for blood vessels in the skin could be a good time to target them with antibodies and prevent malaria infection.

in the liver and exhibit a significant delay in the prepatency period, indicating that CSΔN parasites are impaired in their ability to exit the dermis (*Coppi et al., 2011*). CSΔN sporozoites show only a small reduction in gliding motility in vitro (*Coppi et al., 2011*), suggesting that these mutant sporozoites have additional impairments in vivo. Mutant TRAP-VAL parasites carry mutations in the putative rhomboid-cleavage site of TRAP and similar to the CSΔN mutant, they have a more dramatic reduction in their infectivity after intradermal inoculation, compared to intravenous inoculation (*Ejigiri et al., 2012*). Unlike the CSΔN sporozoites, TRAP-VAL sporozoites display a significantly reduced gliding speed in vitro, moving at approximately 0.5 µm/s in vitro (*Ejigiri et al., 2012*).

Here, we present a quantitative in vivo study on the motility of *Plasmodium* sporozoites over time, and by visualization of dermal vascular endothelia we describe their interaction with dermal blood vessels. We characterize changes in sporozoite motility over the first 2 hr after intradermal inoculation and identify an altered type of sporozoite motility in proximity of blood vessels. We rendered the CSΔN and TRAP-VAL sporozoites fluorescent in order to study the function of these surface proteins in dermal parasite motility and blood vessel recognition and found that CSΔN sporozoites spend more time engaging with blood vessels, yet are unable to enter the blood circulation. Imaging TRAP-VAL sporozoites, we find that their slow gliding speed significantly decreases the volume of tissue explored, which likely results in reduced ability to encounter blood vessels.

## Results

### Sporozoite motility at the dermal inoculation site is increasingly constrained over time

To quantitatively assess sporozoite motility over the first 120 min after inoculation into the skin of a mouse, we generated *P. berghei* sporozoites expressing the fluorescent protein mCherry under the

control of a strong sporozoite-stage promoter (*Figure 1—figure supplement 3*) and visualized them in the ear pinna. 4-min time-lapse stacks were acquired 5 min, 10 min, 20 min, 30 min, 60 min, and 120 min after intradermal inoculation (see *Video 1*) and the paths of migrating sporozoites were manually tracked using Imaris software. Reconstructed tracks were re-centered by plotting to a common origin (*Figure 1A*), which revealed a gradual decrease in parasite dispersal over the first 120 min after inoculation. The mean square displacement (MSD) reflects the dissemination of a motile population from their origin (*Beltman et al., 2009*). To characterize changes in sporozoite dissemination over the first 120 min after inoculation, the MSD of sporozoites was plotted over time. Highest dispersals were seen within the first 15 min after intradermal inoculation (*Figure 1B*). Over the following 110 min, parasite dispersal gradually decreases, which is reflected in a gradual reduction of the slope obtained from linear regression fitting of the MSD plot (*Figure 1B*, inset).

Since the distribution of displacements is more sensitive to the nature of the migration than the MSD, we determined the probability that a sporozoite would reach a given final displacement at different time points after inoculation. This allowed us to ascertain whether sporozoite dissemination mimics that of a Brownian walk. The distribution of this probability P(r) was plotted in a normalized histogram (*Figure 1C*), and it was found that the migration of sporozoites is not well described by an ordinary random walk. In particular, the probability of large displacements is much higher than would be expected for a Brownian walk (*Harris et al., 2012*) (*Figure 1C*). Importantly, the analysis of sporozoite displacement only included sporozoites predominantly moving in linear or meandering patterns.

While the ability of sporozoites to disperse decreases significantly after 20 min and further during the following 100 min, the apparent gliding speed does not change significantly until 60 min after inoculation (*Figure 1D*), which shows that the decrease in MSD is not merely a reflection of sporozoites moving more slowly. This prompted us to look at the pattern with which sporozoites move at the different time points. Over time, an increasing number of parasites engage in continuous circular motility (*Figure 1E*). While at early time points, this sporozoite behavior is seen less frequently, at 20 min and 30 min after inoculation, approximately 22% and 36% of total sporozoites exhibit continuously circling motility. Importantly, at these time points the increased percentage of circling sporozoites is not the result of the other motile sporozoites leaving the field, but rather an increase in the absolute number of sporozoites that are exclusively circling (*Figure 1—figure supplement 1*). However, by 60 and 120 min, there are 20 and 40% fewer sporozoites in the field, respectively, and an increased number of non-motile sporozoites. Thus, at 120 min almost all moving sporozoites are circling (*Figure 1E* and *Figure 1—figure supplement 1*).

Our analysis shows that the change in sporozoite trajectories at 20 and 30 min was not due to their slowing down (*Figure 1D* and *Figure 1—figure supplement 1*). To better understand this increase in parasite confinement, track straightness was calculated. Track straightness is defined as the ratio of displacement to track length (*Beltman et al., 2009*), and smaller ratios reflect more constrained tracks, while straight tracks are characterized by large ratios. This analysis included the entire population of motile sporozoites (meandering sporozoites, as well as sporozoites continuously moving in the same circle). As shown in *Figure 1F*, the straightness of sporozoite tracks gradually decreases over the first 120 min after inoculation with the most significant decrease seen beginning at 20 min after inoculation. These data suggest that sporozoite motility changes over time such that after an initial phase of maximal dispersal, sporozoites move in more constrained circular paths. Of note, a similiar change in sporozoite motility was seen in data acquired from sporozoites imaged after inoculation by a probing infected mosquito, with a significant drop in MSD

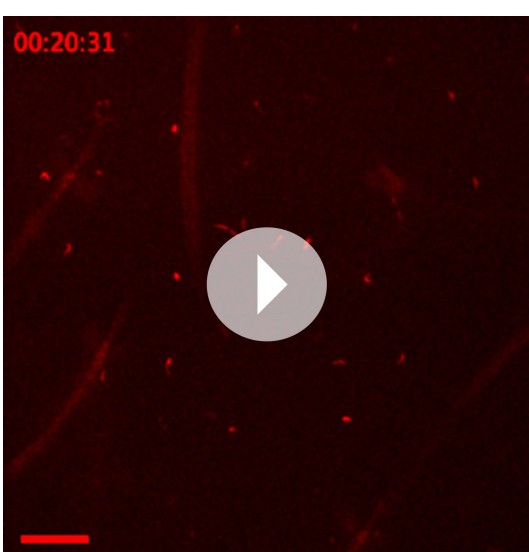

**Video 1.** Time-lapse microscopy of wild-type control *P. berghei* sporozoites from 5 min to 120 min after intradermal inoculation.

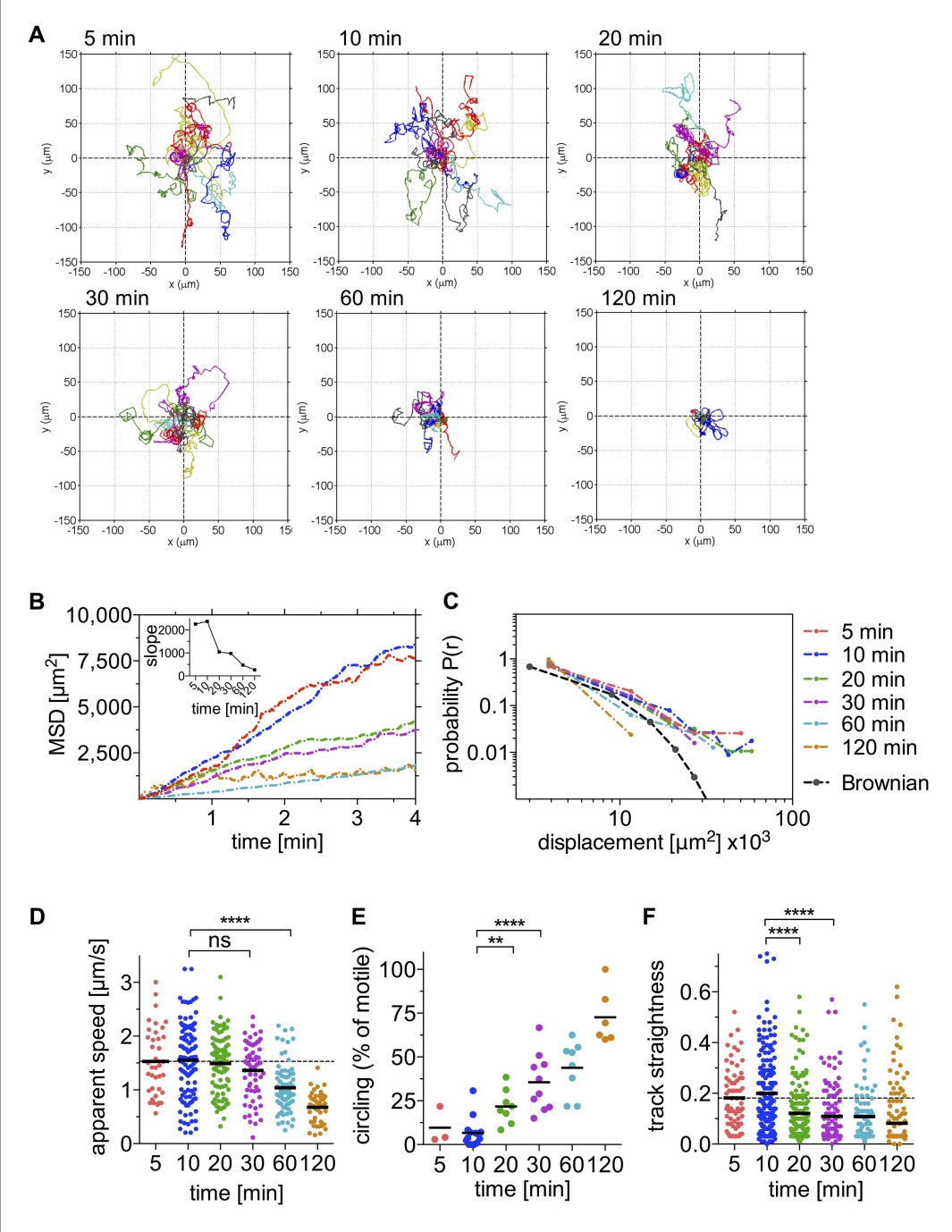

**Figure 1**. Sporozoite motility is increasingly constrained over time and is not well described by a Brownian walk. Time-lapse microscopy of sporozoites was started at the indicated time points after intradermal inoculation and 4 min long videos were acquired. See *Video 1* for a representative time course. (**A**) Meandering and linearly moving sporozoites were manually tracked using Imaris software and for each time point 11 representative reconstructed tracks were plotted to a common origin to visualize parasite dispersal over time. For panels **B**–**F**, a varying number of videos were processed for each time point after inoculation: 5 min (2 videos/37 tracks), 10 min (14 videos/179 tracks), 20 min (7 videos/95 tracks), 30 min (9 videos/129 tracks), 60 min (8 videos/77 tracks), and 120 min (7 videos/67 tracks). (**B**) Mean square displacement (MSD) of sporozoite tracks over the duration of the 4 min video, at indicated time points after inoculation, with inset showing the slope obtained through linear regression fitting of MSD curves. (**C**) The probability distribution P(r) of squared final sporozoite displacements at the end of the 4-min videos. For comparison, the distribution of a Brownian walk is shown. (**D**) Apparent speed of gliding sporozoites. Bars represent

*Figure 1. continued on next page*

*Figure 1. Continued*

mean values and dashed line marks the mean value at 5 min after inoculation. (**E**) The percentage of sporozoites gliding continuously in the same circle throughout the duration of the video, showing an increasing proportion of exclusively circling sporozoites at later time points. Every data point represents one video. (**F**) Straightness of sporozoite tracks, the ratio of displacement to track length of both meandering and linearly moving, as well as continuously circling sporozoites. Bars represent mean values and dashed line marks the mean value at 5 min after inoculation.

The following figure supplements are available for figure 1:

**Figure supplement 1**. Absolute number of sporozoites exhibiting different motility patterns over time.

**Figure supplement 2**. Motility is increasingly constrained over time after sporozoite inoculation by mosquito bite.

**Figure supplement 3**. Generation and verification of a marker-free *Plasmodium berghei* line expressing mCherry under control of the uis4 promoter, using GIMO transfection.

20 min after mosquito bite and a concomitant increase in the number of sporozoites engaging in circular motility (see *Figure 1—figure supplement 2*). Though these data suggest that overall, the pattern of motility is similar whether sporozoites are deposited by mosquito bite or needle; additional experimentation is needed to provide more quantitative data. Unfortunately, these data are more difficult to acquire because of technical issues in finding and capturing, by video-microscopy, sporozoites inoculated by a probing mosquito.

## Sporozoites recognize dermal CD31+ blood vessels

In order to exit the skin and enter the blood circulation, sporozoites must encounter and penetrate blood vessels. Although it was initially postulated that sporozoites might also reach the liver via the lymphatic vessels, a previous study found that sporozoites entering lymphatic vessels in the skin do not travel beyond the draining lymph node, and thus do not participate in liver stage infection (*Amino et al., 2006*; *Radtke et al., 2015*). To characterize the ability of sporozoites to engage with dermal vasculature, we visualized vascular endothelia by labeling the pan-endothelial junction molecule CD31 (also designated as platelet endothelial cell adhesion molecule 1, PECAM-1) by intravenous injection of fluorescently labeled rat anti-CD31 (*Formaglio et al., 2014*). CD31 is a member of the immunoglobulin superfamily and is present on the surface of endothelial cells of both lymphatics and blood vessels, where it is a constituent of intercellular junctions. However, as we performed our experiments within 2–3 hr of intravenous inoculation of anti-CD31, lymphatics were not labeled in our studies, since it takes ~24 hr for intravenously inoculated antibody to get into the extravascular space and taken up by lymphatic vessels (*Sarkisyan et al., 2012*). This was confirmed by labeling lymphatic vessels with antibodies against the lymphatic vessel endothelial receptor 1 (LYVE-1) (*Kilarski et al., 2013*), which in the time frame of our experiments revealed no co-localization with CD31 (*Figure 2—figure supplement 1*). Additionally, lymphatic vessels, being wider and flatter and frequently possessing blind endings, have a different shape than blood vessels.

We observed sporozoites gliding in close proximity to CD31-labeled vessels and frequently found sporozoites gliding in parallel to or in circular paths around the vessel (*Figure 2A* and *Video 2*). Surprisingly, sporozoites do not immediately penetrate blood vessels after contacting them. Instead, they move on and around vessels for significant periods of time, frequently even detaching from vessels and moving back into avascular areas of the tissue (*Figure 2A* and *Video 2*). We quantified the time sporozoites spend in proximity to blood vessels and found that on average, sporozoites spend 28% of their gliding time near or on vessels, which in our 4-min videos correspond to over a minute of total gliding time.

To determine whether sporozoite motility changes with the encounter of a blood vessel, we manually tracked sporozoite trajectories and produced two data sets; the first for time periods in which sporozoites move in proximity to CD31-labeled structures (within ~3 µm of the vessels) and the second for time periods of gliding in areas without CD31-labeled vessels. A representative maximum intensity projection of this tracking is shown in *Figure 2A* with the two data sets shown in different colors. Analysis of the two tracking sets showed that compared to sporozoites gliding independently of CD31-labeled

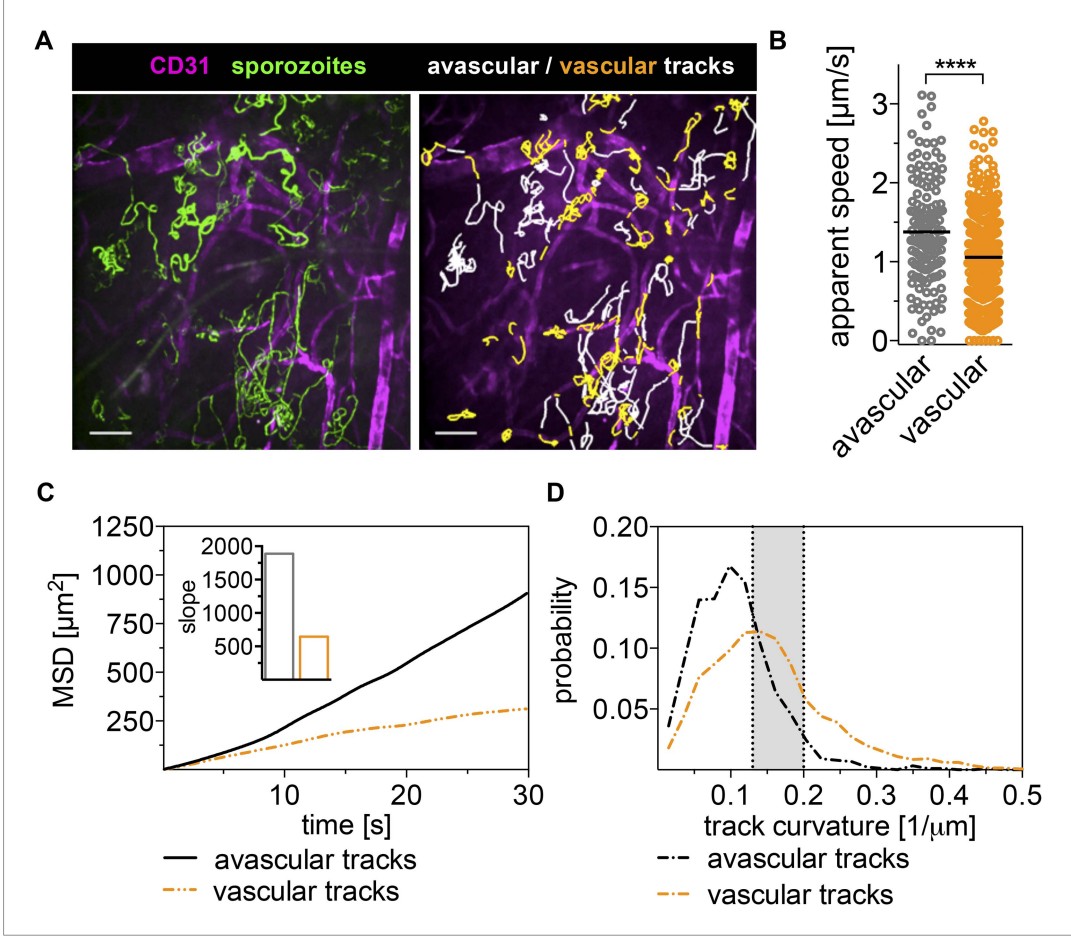

**Figure 2**. Sporozoites recognize dermal blood vessels. Time-lapse microscopy of sporozoites, 10 min after intradermal inoculation, and CD31-labeled vascular endothelia. (**A**) Maximum intensity projection over 240 s shows trajectories of moving sporozoites (green) and blood vessels (far red) (see *Video 2*). Sporozoites were manually tracked using Imaris software and tracks overlapping with CD31+ vessels are colored orange, while tracks in between CD31+ vessels are white. Scale bar, 25 µm. For panels **B–D**, 18 videos were processed and 313 tracks of sporozoites moving in proximity of blood vessels and 163 tracks of sporozoites moving in between blood vessels were generated. (**B**) Apparent speed, with lines representing mean values, reveals a decrease in gliding speed once sporozoites are in proximity of blood vessels. (**C**) MSD plot shows significant decrease in displacement of sporozoite tracks in the vicinity of vasculature. Inset shows decreased slope of perivascular tracks obtained from linear regression fitting ($R^2$ values: $R_{perivascular} = 0.9884$; $R_{avascular} = 0.9924$). (**D**) Extrapolated track curvature, as defined by 1/radius (where the radius is measured in µm), of perivascular and avascular sporozoite tracks. Kolmogorov–Smirnov two-sample test was performed and found difference of perivascular and avascular curvature highly significant ($p = 10^{-57}$). Shaded area shows range of curvature of dermal capillaries (*Yen and Braverman, 1976*; *Braverman, 1997*).

The following figure supplement is available for figure 2:

**Figure supplement 1**. Anti-CD31 does not stain lymphatic vessels early after its inoculation.

---

vessels, the average apparent speed of gliding sporozoites decreases once they approach a blood vessel from 1.38 µm/s to 1.06 µm/s, and this difference is highly significant (p value < 0.0001) (*Figure 2B*).

To compare the dispersal of sporozoites gliding in avascular tissue to those gliding in proximity to vascular structures, the MSD of the two populations was plotted over time. To normalize for track length in this analysis, all tracks were fragmented to 30-s intervals. As shown in *Figure 2C*, the displacement of sporozoites gliding on or in proximity to blood vessels is decreased, as is the slope obtained from linear regression fitting of the MSD plot (*Figure 2C*, inset). This is partially a result of the decrease in gliding speed of these sporozoites, but also due to an increase in sporozoite

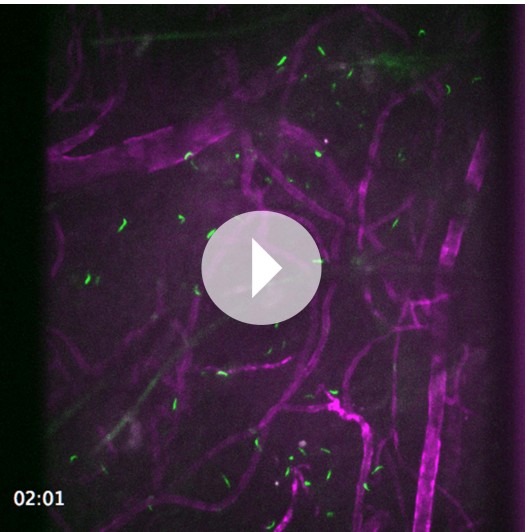

02:01

**Video 2.** Wild-type control *P. berghei* sporozoites 10 min after intradermal inoculation together with CD31-labeled vascular endothelia. Maximum projection shown in *Figure 2*.

confinement. This is demonstrated by an analysis of the curvature of sporozoite tracks, which is the curvature of the circles fitted to every 10-s interval of a given track and is defined as the inverse of the radius of that circle (see https://github.com/kevink-chiou/Malaria-Motility and 'Materials and methods'). The analysis revealed that while avascular tracks have little curvature, most commonly 0.077–0.125 $\mu m^{-1}$, the majority of sporozoite tracks that overlap with blood vessels describe trajectories that have a curvature of 0.125–0.2 $\mu m^{-1}$, which, remarkably, equals the curvature of the circumference of dermal capillaries (*Figure 2D*) (*Yen and Braverman, 1976*; *Braverman, 1997*). These data demonstrate that sporozoite motility can change in response to environmental signals encountered by the sporozoite and suggest that sporozoites in the dermis exhibit two types of motility: 'avascular motility', which is defined by high speed and less confinement, leading to maximal displacements in avascular areas and a more constrained 'perivascular motility', which is observed once the parasite glides in proximity of a blood vessel.

## Motility of CSΔN and TRAP-VAL mutants in the dermis

Recent work suggested that both CSP and TRAP have important roles for exit from the dermis with the demonstration that CSΔN parasites, which express a truncated form of CSP, and TRAP-VAL sporozoites, which carry mutations in the putative rhomboid-cleavage site of TRAP, have a more dramatic reduction in their infectivity after intradermal inoculation, compared to intravenous injection (*Coppi et al., 2011*; *Ejigiri et al., 2012*). We characterized the motility of CSΔN and TRAP-VAL sporozoites in the dermis to elucidate the function of these surface proteins in vivo, specifically focusing on parasite motility and blood vessel entry. Both mutants were generated in a drug selection cassette-free parental *P. berghei* parasite line expressing mCherry under the control of a strong sporozoite-stage promoter (*Figure 1—figure supplement 3*; *Figure 3—figure supplements 1, 2*). CSΔN and TRAP-VAL sporozoites were imaged 10 and 30 min after intradermal injection into the ear pinna of an anesthetized mouse, and sporozoite trajectories were visualized by maximum intensity projection of the 4-min videos (*Figure 3A*, *Video 3* and *Video 4*). To quantify the proportion of mutant sporozoites that were motile, sporozoites were manually counted, with sporozoites being considered non-motile if they moved less than approximately 2 μm throughout the duration of the video (*Figure 3B*). An average of 66% of wild-type control sporozoites are motile 10 min after intradermal injection and 52% are motile at the 30 min time point. At 10 min after injection, only 47% of CSΔN sporozoites are motile, however, this number is stable over time and similar to wild-type control levels at 30 min. The analysis of TRAP-VAL sporozoites showed that only 40% and 20% of the inoculated sporozoites are motile at 10 min and 30 min after inoculation, respectively (*Figure 3B*).

Average speed of sporozoites was determined by manually tracking sporozoites throughout the duration of video, which includes periods of stopping and phases of continuous movement. Compared to wild-type control sporozoites, which move with an average of 1.57 μm/s, CSΔN sporozoites move with 26%-reduced average speed of 1.14 μm/s (*Figure 3C*). In contrast, at 10 min after inoculation, TRAP-VAL sporozoites glide much more slowly, with an average speed of 0.72 μm/s, which is a reduction of 54% compared to the wild-type control (*Figure 3C*) and this is further reduced to 0.42 μm/s at 30 min after inoculation.

In agreement with the reduction in gliding speed of the CSΔN and TRAP-VAL sporozoites in vivo, it became evident that 10 min after intradermal inoculation, both mutant sporozoites disperse less than wild-type control parasites, which was visualized by plotting reconstructed sporozoite trajectories to a common origin (*Figure 4A*). To quantitatively characterize this difference in the motility pattern,

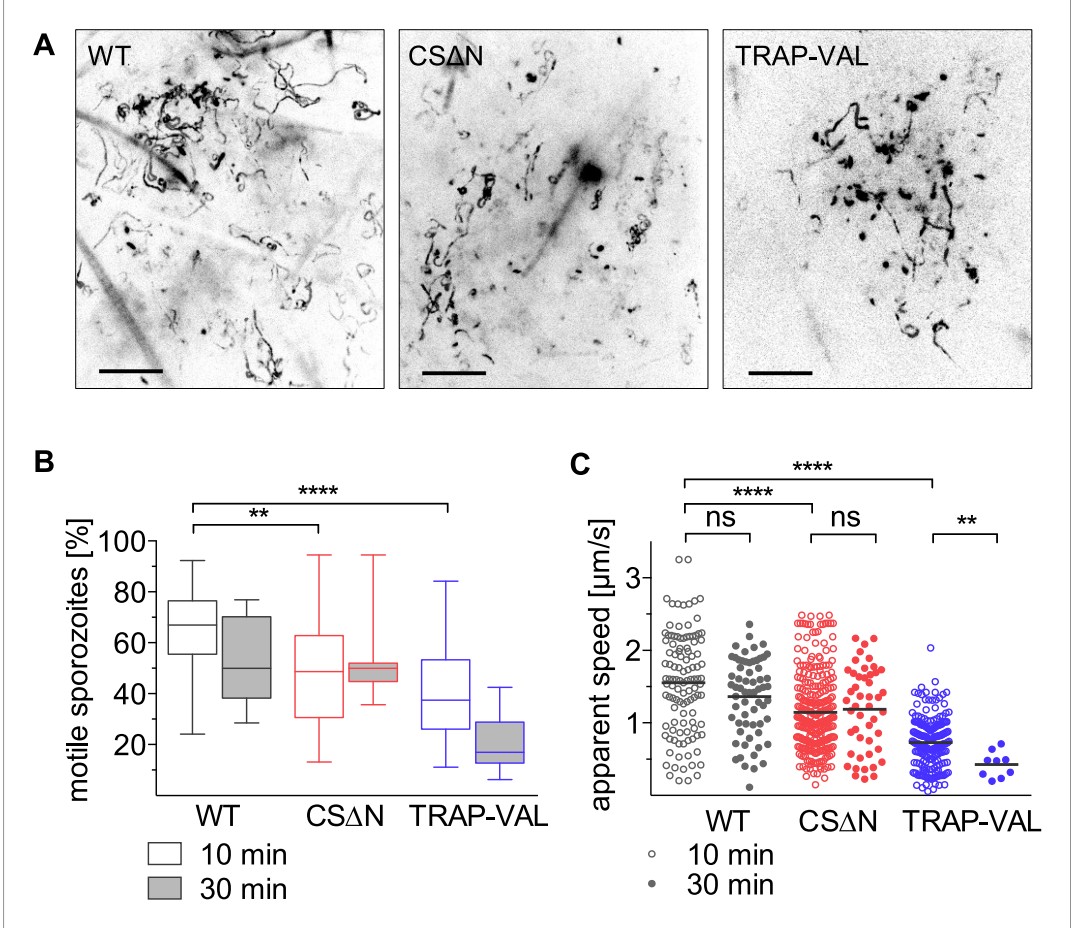

**Figure 3.** Intradermal motility of CSΔN and TRAP-VAL sporozoites. Time-lapse microscopy of CSΔN and TRAP-VAL sporozoites (see **Video 3** and **Video 4**). (**A**) Maximum intensity projections of videos acquired 10 min after inoculation visualize trajectories of sporozoites. Scale bars, 50 μm. (**B**) Proportion of motile sporozoites was manually counted in videos acquired 10 min (WT, 33 videos; CSΔN, 31 videos; TRAP-VAL, 43 videos) and 30 min (WT, 8 videos; CSΔN, 11 videos; TRAP-VAL, 10 videos) after inoculation. (**C**) Apparent speed of sporozoites 10 min and 30 min after inoculation, averaged over the 4-min video.

The following figure supplements are available for figure 3:

**Figure supplement 1.** Generation of mutant CSΔN parasites in the marker-free *P. berghei* line expressing mCherry under control of the uis4 promoter.

**Figure supplement 2.** Generation of mutant TRAP-VAL parasites in the marker-free *P. berghei* line expressing mCherry under control of the uis4 promoter.

sporozoites gliding 10 min after intradermal inoculation were tracked and the MSD was plotted over time. The visibly reduced dispersal of mutant sporozoites was reflected in a decreased slope obtained from linear regression fitting of the MSD plot (**Figure 4B**, inset). Compared to wild-type control sporozoites, slope$_{WT}$ = 2370, the slope of mutant sporozoites is reduced to slope$_{CSΔN}$ = 1190 and slope$_{TRAP-VAL}$ = 740, resulting in a decrease of over 40% and 70% of final displacement of CSΔN and TRAP-VAL sporozoites, respectively.

## CSΔN and TRAP-VAL sporozoites contact dermal vascular endothelia

Visualization of sporozoites and dermal blood vessels shows that sporozoites glide alongside dermal vascular endothelia and frequently circle extensively around the vessel (as seen in **Figure 2A** and

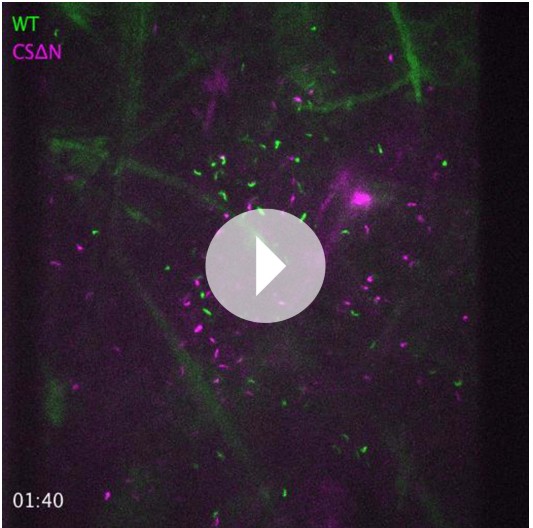

**Video 3.** Wild-type control (green) and CSΔN (far red) sporozoites 10 min after intradermal inoculation. Note that an overlay of two independently acquired videos was generated using ImageJ to allow direct comparison of WT to CSΔN sporozoite motility. Individual maximum projections shown in *Figure 3*.

**Video 4.** Wild-type control (green) and TRAP-VAL (far red) sporozoites 10 min after intradermal inoculation. Note that an overlay of two independently acquired videos was generated using ImageJ to allow direct comparison of WT to TRAP-VAL sporozoite motility. Individual maximum projections shown in *Figure 3*.

*Video 2*). Additionally, sporozoites that are moving in proximity to blood vessels often disengage and move away from the blood vessel into avascular tissue. To quantitatively assess the time a population of sporozoites spends gliding alongside or around blood vessels, wild-type control and mutant sporozoites were manually tracked while moving in proximity to blood vessels (*Video 2*, *Video 5* and *Video 6*). Unexpectedly, it was found that compared to wild-type control sporozoites, which spend 28% of their motile time gliding in close proximity to blood vessels, CSΔN sporozoites spend significantly more time in association with dermal vascular endothelia; an average of 40.7% of their motile time is spent close to blood vessels (*Figure 5B*, left panel). In contrast, TRAP-VAL sporozoites spend the same proportion of their motile time gliding close to blood vessels as wild-type control parasites (*Figure 5B*, left panel).

To determine whether the mutant sporozoites undergo a similar switch from 'avascular motility' to 'perivascular motility' when approaching a blood vessel, we manually tracked mutant sporozoites and produced two data sets, as described above, for sporozoites moving in proximity to CD31-labeled structures and for sporozoites gliding in areas without CD31-labeled vessels. MSD analysis of the two tracking sets revealed that similar to wild-type control sporozoites, perivascular CSΔN tracks have a decreased MSD and gliding speed (*Figure 5C,D*), indicating that while CSΔN sporozoites spend more time gliding in proximity to vasculature, the switch from 'avascular' to 'perivascular motility' appears normal. TRAP-VAL sporozoites also exhibit differential 'avascular' and 'perivascular motility', although the differences in MSD and speed between these two types of tracks is less pronounced (*Figure 5C,D*).

## Decreased blood vessel exit of CSΔN and TRAP-VAL sporozoites
Intravital microscopy of mutant CSΔN and TRAP-VAL sporozoites in conjunction with fluorescently labeled dermal vascular endothelia was performed, and invasion events were manually quantified (*Figure 5E*, *Video 7* and *Video 8*). Invasion was classified as previously described (*Amino et al., 2006*), with blood vessel invasion defined by a sudden increase in speed or visual entry of the blood vessel and disappearance out of the field of view. It should be noted that the speed with which sporozoites appear to be carried out of the field is slower in videos that include CD31-labeling, as a result of the slower acquisition speed due to capture of two channels. Lymphatic invasion was

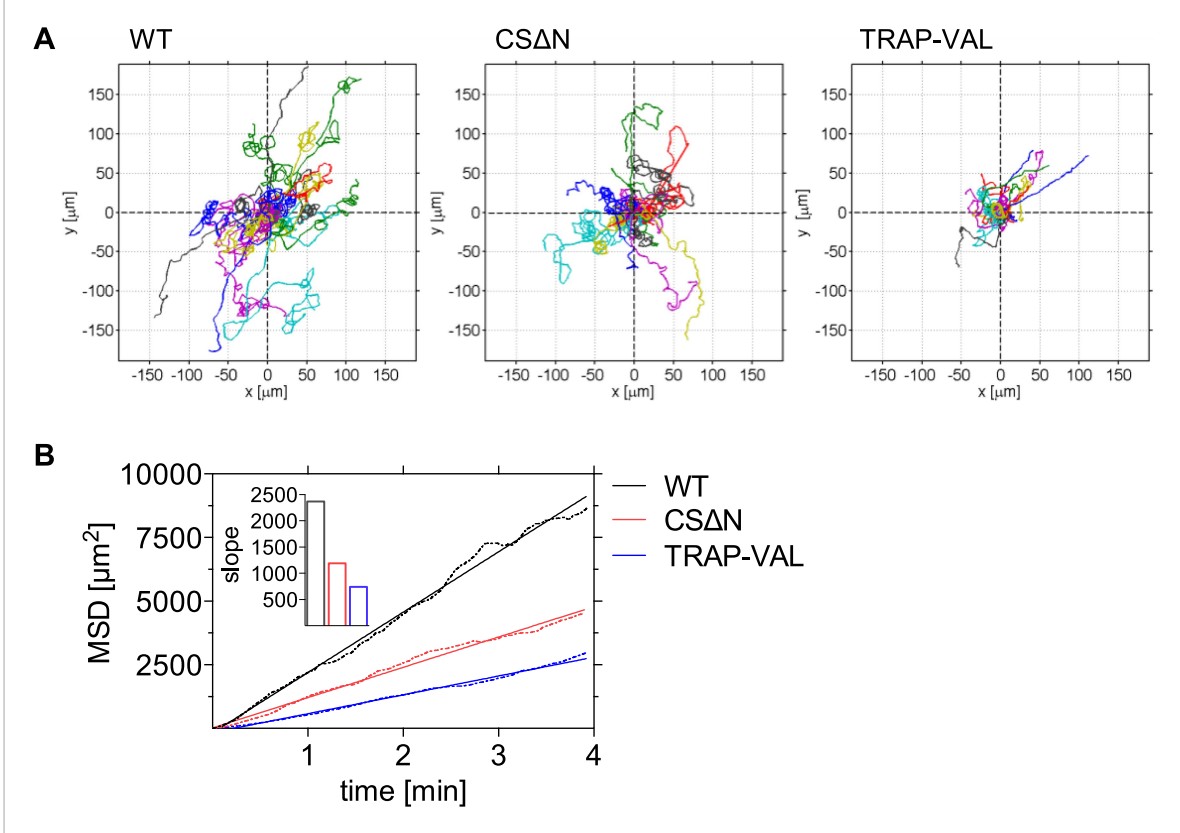

**Figure 4**. Decreased dispersal of CSΔN and TRAP-VAL sporozoites in vivo. Time-lapse microscopy of CSΔN and TRAP-VAL sporozoites. (**A**) Sporozoites were manually tracked using Imaris software and 20 representative tracks of each WT, CSΔN, and TRAP-VAL sporozoites were plotted to a common origin, revealing a decreased dispersal of mutant sporozoites. (**B**) MSD plot over time. Data shown was obtained from tracking 179 WT sporozoites (12 videos), 211 CSΔN sporozoites (29 videos), and 215 TRAP-VAL sporozoites (33 videos). Inset shows the decreased slope of mutant parasites obtained from linear regression fitting ($R^2$ values: $R_{WT}$ = 0.9926; $R_{CS\Delta N}$ = 0.9890; $R_{TRAP-VAL}$ = 0.9932).

defined by the switch from directed forward movement to sideward drifting with a low velocity. On average, in a 4-min video beginning at 10 min after inoculation, 2.38% of wild-type control sporozoites in the field of view enters the blood circulation, and 2.06% of sporozoites enters the lymphatic system (**Figure 5E**). Blood vessel invasion of the CSΔN mutant is reduced to 0.23% and only 0.05% of TRAP-VAL sporozoites is seen to enter into blood vessels (**Figure 5E**), which is consistent with the decrease in infectivity of these mutants upon intradermal inoculation (**Coppi et al., 2011**; **Ejigiri et al., 2012**). Invasion of the lymphatic system by CSΔN sporozoites is reduced to 0.26% and not detected at all in the case of TRAP-VAL sporozoites (**Figure 5E**). Importantly, CD31-labeling is not interfering with entry into dermal blood or lymphatic vessels or parasite dispersal, as vessel invasion rates, as well as sporozoite MSDs, are comparable in videos acquired in mice with and without labeled blood vessels (**Figure 5—figure supplement 1**).

This dramatic decrease in blood vessel invasion by the mutants is likely due to several factors. As shown in **Figure 4**, these mutants sample less of their environment; however, this is unlikely to fully explain their dramatically decreased entry into the blood and lymphatic circulations. Since only motile sporozoites are included in the calculation of MSD, it is important to consider the increased number of non-motile CSΔN and TRAP-VAL sporozoites that together with the decreased dispersal, could account for the low numbers of mutants entering the blood circulation. To look at this, we normalized the total time control and mutant sporozoites spent in proximity of blood vessels, by including all sporozoites in the field, that is, both motile and non-motile sporozoites. As shown in **Figure 5B** (right panel), TRAP-VAL sporozoites spend on average, 40% less time on vessels compared to wild-type control sporozoites, a decrease that is due, in part, to the large number of non-motile TRAP-VAL

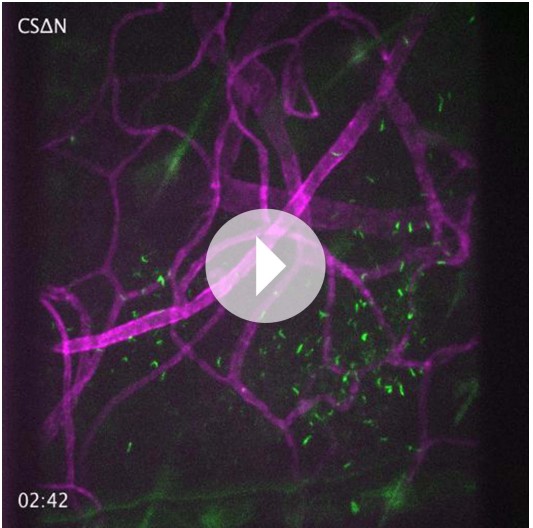

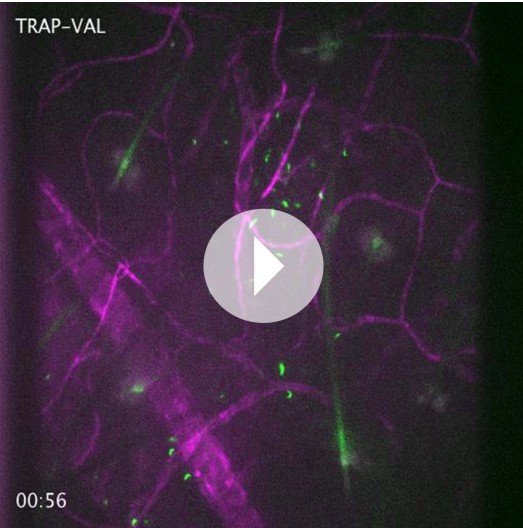

**Video 5.** CSΔN sporozoites 10 min after intradermal inoculation together with CD31-labeled vascular endothelia. Maximum projection shown in *Figure 4*.

**Video 6.** TRAP-VAL sporozoites 10 min after intradermal inoculation together with CD31-labeled vascular endothelia. Maximum projection shown in *Figure 4*.

sporozoites. Nonetheless, the population of TRAP-VAL sporozoites still spends approximately 10% of its time on or near blood vessels, yet blood vessel entry events were reduced by 97%, suggesting that even when TRAP-VAL sporozoites do contact vessels they cannot invade them. This is even more dramatically illustrated by the CSΔN mutant that spends approximately the same total time on vessels as the control wild-type sporozoites (*Figure 5B*, right panel), yet blood vessel entry events were reduced by 90%. These data point to an additional impairment in blood vessel entry, that is, even when they do contact blood vessels, they cannot invade them. Indeed, both CSΔN and TRAP-VAL mutations result in the increased surface exposure of cell-adhesion domains: in CSΔN sporozoites, the absence of the N-terminus leads to constitutive exposure of the cell-adhesive type I thrombospondin domain in the C-terminus of the protein (*Coppi et al., 2011*) and in TRAP-VAL sporozoites, the altered proteolytic cleavage site leads to the retention of TRAP molecules on the sporozoite's surface (*Ejigiri et al., 2012*). Thus, the inability of these mutants to regulate their adhesive interactions likely results in their inability to complete the entry process.

## Discussion

The gliding motility of *Plasmodium* sporozoites is essential for their infectivity. Previous reports have observed sporozoite motility in the dermis (*Amino et al., 2006*; *Hellmann et al., 2011*), but an in-depth analysis of their motility over time has not been performed. Intravital imaging of sporozoites over the first 120 min after inoculation highlights significant changes in their motility in this time frame. Within the first 20 min after intradermal inoculation, sporozoites displace with high speed and minimal constraint, which likely maximizes the volume of tissue that is explored by the population of sporozoites. After this initial period of dispersal, sporozoite motility changes. Most significantly starting 20 min after inoculation, the migratory pattern of sporozoites becomes increasingly confined, and a growing number of sporozoites engage in continuous circular motility. We hypothesize that this motility pattern optimizes engagement with blood vessels that will ultimately lead to blood vessel entry. 1 hr after inoculation, sporozoites begin to slow down and a significant proportion stop moving with the proportion of non-motile sporozoites increasing to 70% by 2 hr. Importantly, this late-phase slow down is distinct from the change in trajectory that occurs 20 and 30 min post inoculation when the speed and proportion of non-motile sporozoites does not change from what is observed at earlier time points. The reasons for this late-phase slow down are not known, but possibilities include resource exhaustion or encountering a host cell or factor that leads to their arrest.

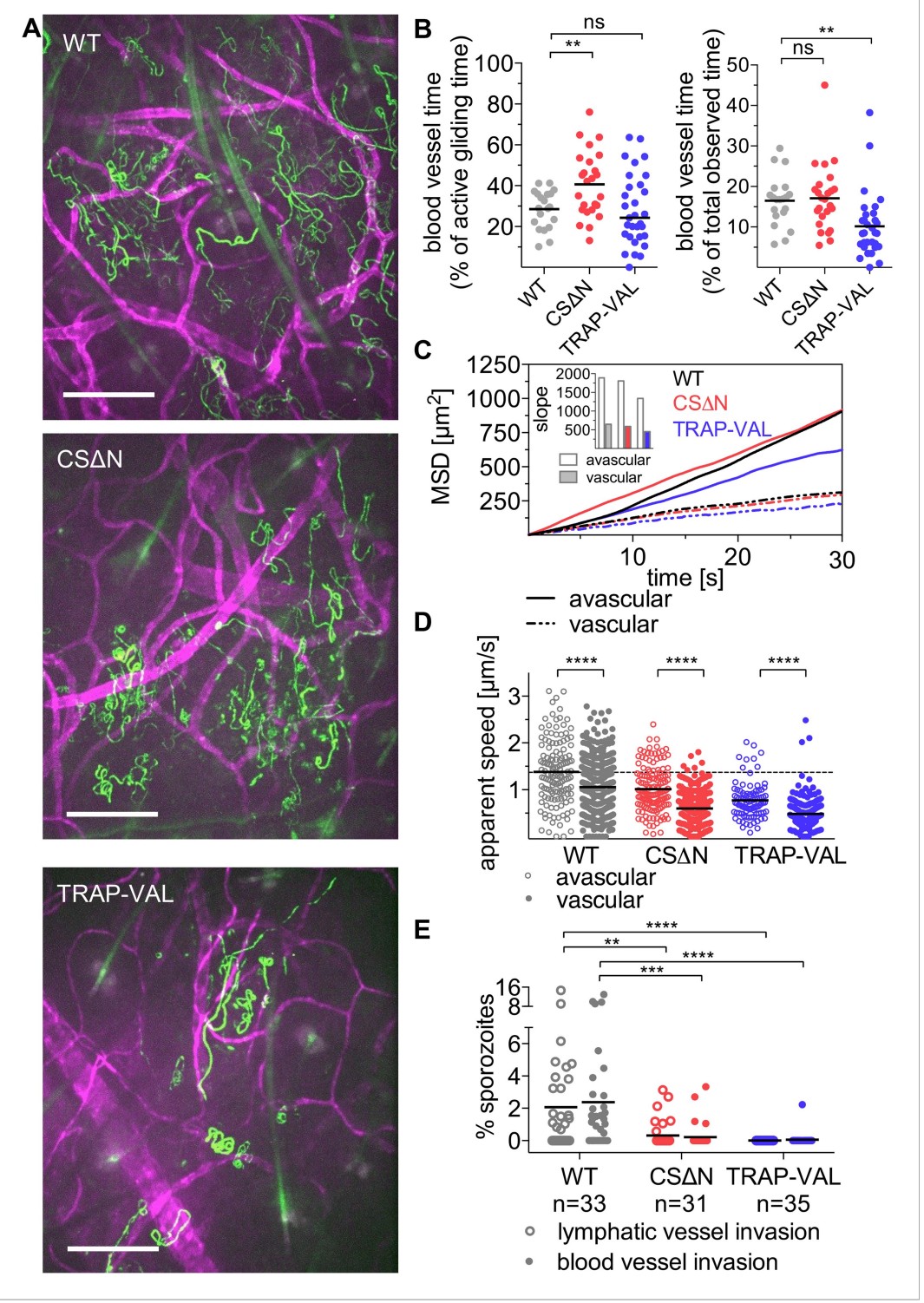

**Figure 5**. Sporozoite interaction with dermal vascular endothelia. Time-lapse microscopy of CSΔN and TRAP-VAL sporozoites 10 min after intradermal inoculation (see *Video 2*, *Video 5*, and *Video 6*). (**A**) Maximum intensity projection over 240 s reveals trajectories of WT, CSΔN, and TRAP-VAL sporozoites (green) moving in the dermis with CD31-labeled vascular endothelia (far red). Scale bars, 50 µm. For panels **B**–**D**, every data point represents one video and 18 WT, 25 CSΔN, and 24 TRAP-VAL videos were processed. (**B**) The time sporozoites spend in proximity to CD31-labeled vessels was quantified by manual tracking of sporozoites and is expressed as percent of motile time (left panel) or total observed time (right panel). Graphs show significant increase in the time CSΔN, but not

*Figure 5. Continued*

TRAP-VAL sporozoites spend actively gliding on or around blood vessels. (**C**) To generate the MSD of sporozoites moving in proximity to CD31-labeled vessels, 313 WT, 352 CSΔN, and 192 TRAP-VAL tracks of sporozoites moving in proximity to CD31-labeled vessels were generated and compared to 163 WT, 121 CSΔN, and 89 TRAP-VAL tracks of sporozoites moving in avascular areas of the dermis. To calculate the MSD of the obtained sporozoite tracks with various duration times, all tracks were uniformly fragmented into tracks of 30 s in duration. MSD plot shows decrease in displacement of mutant sporozoite tracks in proximity to blood vessels. Inset shows slopes of linear regression fitting (all $R^2$ values ≥0.988). (**D**) Apparent speed of wild-type control and mutant sporozoites in the proximity of blood vessels or in avascular portions of the dermis with lines representing mean values. Dashed line marks the mean value of wild-type control in avascular portions of the dermis. These data reveal a significant decrease in gliding speed of CSΔN and TRAP-VAL sporozoites, once the sporozoite is approaching vasculature, similar to wild-type control sporozoites. (**E**) Blood and lymphatic vessel invasion events were manually counted. Invasion was classified similar to previously descriptions (*Amino et al., 2006*), with blood vessel invasion being defined by a sudden increase in speed or visual entry of the blood vessel and disappearance out of the field of view. Lymphatic invasion was characterized by the switch from directed forward movement to sideward drifting with a low velocity (see *Videos 7* and *Video 8*). The proportion of invading sporozoites as percent of total sporozoites in the field of view is shown. In the duration of the acquired 4 min videos, 2.38% and 2.06% of WT sporozoites invaded blood vessels and lymphatic vessels, respectively. This was in sharp contrast to the blood and lymphatic vessel invasion rates of CSΔN (0.23% and 0.26%) and TRAP-VAL (0.05% and 0%); n equals number of videos analyzed.
The following figure supplement is available for figure 5:

**Figure supplement 1**. Blood vessel and lymphatic vessel invasion events and sporozoite MSD in videos acquired with and without CD31-labeling of dermal vasculature.

In contrast to previous reports of apparently random motility of sporozoites in the dermis, which was suggested to be only environmentally constrained (*Amino et al., 2006*; *Hellmann et al., 2011*), our data show that distribution of displacements of migrating sporozoites is not Gaussian, which would be expected if sporozoites migrated in a pattern similar to a Brownian walk. Our observations of an altered sporozoite migration in proximity to blood vessels point to recognition of the vessel by the sporozoites, further demonstrating that sporozoite migration in the dermis is not well described by the statistics of a Brownian walk. This suggests that internal components or external host–pathogen interactions bias the directionality of sporozoite movement away from Brownian motion. The larger than expected displacements may allow the sporozoite to cover large distances between the blood vessels, which may enable the sporozoite to encounter blood vessels with higher efficiency than Brownian random walkers. We found that once the sporozoite approaches a blood vessel, the more linear type of motility is lost and the sporozoite switches to a more curved and confined motility. This points to a specific host–parasite interaction between the parasite and the environment of the blood vessel and provides evidence for the ability of the sporozoite to alter motility in response to host factors. Indeed, in vitro, sporozoites have been shown to respond to obstacle cues in micro-fabricated arrays so it is possible that environmental constraints alone account for the switch in the sporozoites' migration pattern (*Hellmann et al., 2011*). However, our observation that the sporozoites' walk statistics change in proximity to blood vessels indicates that there is a specific recognition event that leads to this change.

Sporozoites appear to engage in two types of motility: 'avascular motility' which is likely to optimize dispersal, and 'perivascular motility' defined by more confined trajectories with increased curvature. Our analysis of track curvature reveals that sporozoites gliding in the proximity of blood vessels describe trajectories with local curvatures that interestingly relate to the curvature of the capillary vessel (*Yen and Braverman, 1976*; *Braverman, 1997*). This suggests that 'perivascular motility' enhances contact with capillaries, rather than the larger dermal terminal arterioles or post-capillary venules (*Yen and Braverman, 1976*; *Braverman, 1997*). Indeed, it has been observed previously, that the curvature of the sporozoite matches the geometry of capillaries (*Vanderberg, 1974*) and more recently, a study using stochastic computer simulations of sporozoite motility found that in these models, sporozoite interaction with capillaries is greatly enhanced if the parasite has the ability to adapt its curvature and torsion to the surrounding environment (*Battista et al., 2014*). In addition to the change in sporozoite trajectories, we found that they also had a significant decrease in

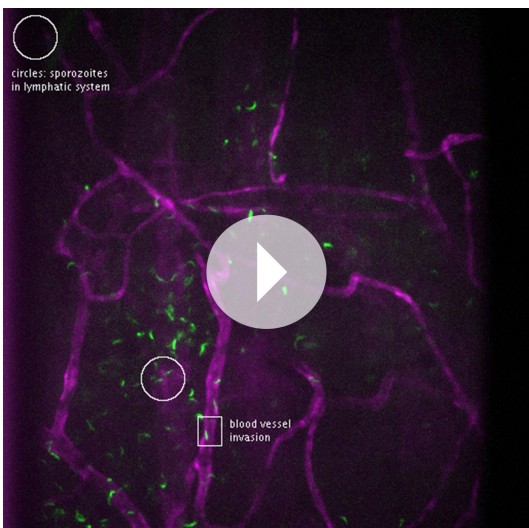 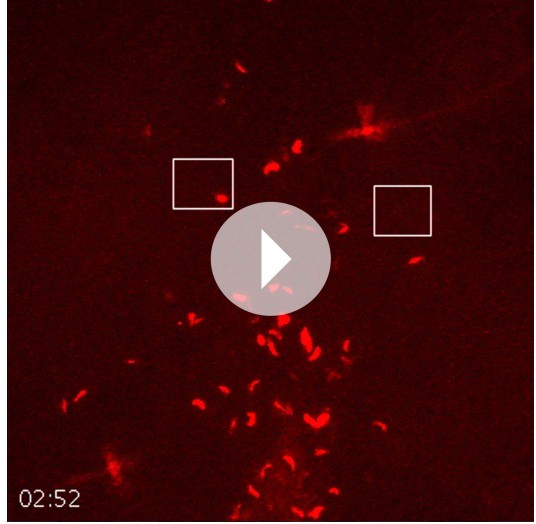

**Video 7.** Wild-type control sporozoites 10 min after intradermal inoculation together with CD31-labeled vascular endothelia highlighting blood and lymphatic vessel invasion events. Squares indicate blood vessel invasion and circles point to sporozoites entered into the lymphatic system. Blood vessel invasion events are characterized by a sudden increase in speed (square on the left) or visual entry of the blood vessel and disappearance out of the field of view (square on the right). Lymphatic invasion was characterized by the switch from directed forward movement to sideward drifting with a low velocity.

**Video 8.** Blood vessel invasion by wild-type control sporozoites in the absence of CD31 labeling. Imaging was performed 10 min after intradermal inoculation and squares indicate blood vessel invasion events, characterized by a sudden increase in speed.

speed when in the vicinity of blood vessels. These data build upon and solidify the previous observation of Amino et al., who quantified the speed of a small number of sporozoites gliding around blood vessels and reported a decrease after contact with the vessel (*Amino et al., 2006*).

Intravital imaging of mutant sporozoites that do not efficiently exit the dermis provides additional insights into the factors required for dermal exit. The importance of high sporozoite gliding speed is highlighted by the behavior of the TRAP-VAL mutant that moves with less than half the motile speed of wild-type control parasites in the dermis. This decrease in speed corresponds to the reduction of final MSD, which reduces the area of tissue explored by 70%. Consequently, TRAP-VAL sporozoites have a significantly decreased chance to encounter and invade a blood vessel, which together with the increase in number of non-motile TRAP-VAL sporozoites, largely explains their reduced infectivity (*Ejigiri et al., 2012*). Although there is no evidence for an innate host immune response that may impair the exit of sporozoites from the dermis, the observed rapid dispersal of wild-type control sporozoites suggests that the parasite has evolved to leave the skin quickly, possibly as a means to escape from phagocytic host cells (*Amino et al., 2008*).

CSΔN sporozoites engage in a pattern of 'avascular' and 'perivascular motility' comparable to that of wild-type sporozoites, suggesting that while gliding in avascular parts of the dermis, their motility is normal. Of particular interest was the observed increase in the association of CSΔN sporozoites with dermal vascular endothelia. Since 'perivascular motility' is more constrained, this, together with the somewhat decreased gliding speed, explains the overall decrease in dispersal of this mutant. It was previously proposed that the function of the N-terminal portion of CSP is to mask the cell-adhesive TSR domain as sporozoites migrate through the skin and controlled proteolytic processing of CSP by a parasite protease regulates the exposure of the TSR (*Coppi et al., 2005*, *2011*). We find that the constitutively exposed TSR on the surface of CSΔN sporozoites leads to increased time spent on vascular endothelia. We therefore hypothesize that the ability of the sporozoite to dynamically regulate exposure of the TSR is necessary for continued migration through the dermis, which may be crucial for the sporozoite to locate the areas of the vasculature that allow for blood vessel entry or for blood vessel entry itself. More work is needed to determine if this is the case, however, our data with

both TRAP-VAL and CSΔN mutant sporozoites demonstrate that the sporozoite's ability to regulate its adhesive intereactions is a key factor for successful movement through and out of the dermis.

We show that wild-type control sporozoites spend over a quarter of their time engaging with blood vessels, frequently without successful entry into the blood circulation. This suggests that only certain areas of the vasculature are suitable for sporozoite entry. It is known that the abluminal surface of dermal capillaries is not uniform; for example, the basement membrane surrounding the capillary bed is not continuous and through its gaps, pericytes form tight junctions with the underlying endothelial cell (*Braverman, 1997*). Thus, there may be certain areas of the vasculature that are preferentially invaded by sporozoites.

With this study, we refine our understanding on how the sporozoite navigates through and out of the dermis. Mutant sporozoites that display modest changes in motility pattern or speed have dramatic decreases in blood vessel invasion and infectivity. These data suggest that exit from the inoculation site presents an important barrier for the sporozoite and is therefore an opportune time to intervene. Indeed, the partial success of RTS,S, the only malaria vaccine candidate in Phase III clinical trials, may result from antibodies that target sporozoites in the dermis (*Foquet et al., 2014*). As we build upon RTS,S, it will be important to consider expanding the antigen repertoire to include proteins critical for dermal exit. Additionally, our data suggest that an analysis of dermal sporozoite motility could be incorporated in the evaluation of future vaccine strategies to predict efficiency of new vaccine targets.

## Materials and methods

### Mosquito infection

*Anopheles stephensi* mosquitoes were fed on Swiss Webster mice infected with *P. berghei* ANKA wild-type control, CSΔN, or TRAP-VAL parasites as previously outlined (*Coppi et al., 2011*). Salivary glands were dissected 18–22 days after the infectious blood meal, homogenized in DMEM, and sporozoites were counted using a hemocytometer.

### Generation of drug selection marker-free PbmCherry-expressing parasites

The *P. berghei* ANKA line 1596cl1 was used for transfection with a construct which targets the *230p* locus (PBANKA_030600) and inserts mCherry, thereby removing the selectable marker consisting of human dihydrofolate reductase and the yeast cytosine deaminase and uridyl phosphoribosyl transferase (h*dhfr::yfcu*), according to the GIMO (gene insertion/marker out) technique, which has previously been described (*Lin et al., 2011*). The transfection vector, which lacks a drug selectable marker cassette, was obtained using the targeting construct pL1628 (*Lin et al., 2011*). The 5′- and 3′-regulatory elements were replaced by the 5′- and 3′-UTR of *uis4* (*Mueller et al., 2005*), which places mCherry under the control of this sporozoite and liver stage promoter in the new transfection construct pL1937. Transfection of 1596cl1 was performed using standard transfection methods (*Janse et al., 2006*), and after transfection, negative selection was applied by treating mice with the drug 5-FC in drinking water at 1 mg/ml (*Orr et al., 2012*) for four consecutive days, starting 24 hr after transfection. The resulting drug selection marker-free transfected line (2204) was cloned by limiting dilution in mice and correct integration of mCherry into the *230p* locus was demonstrated by PCR analysis using primers 5510 and 4958 to amplify the 5′ integration site, and primers 5515 and 5511 to amplify the 3′ integration site (*Figure 1—figure supplement 3*). The absence of the h*dhfr::yfcu* selection cassette was verified by PCR with primers 4698 and 4699 (see *Supplementary file 1* for primer sequences).

### Generation and verification of PbmCherry-CSΔN and TRAP-VAL parasites

#### Transfection

The drug selection cassette-free parasite line (*Pb*ANKA-Cherry 2204cl5), referred to as 'wild-type control', was used for transfection with CSΔN and TRAP-VAL constructs which have previously been described (*Coppi et al., 2011*; *Ejigiri et al., 2012*). Thus, mutant versions of *CSP* and *TRAP* with their endogenous control elements, as well as the h*dhfr* selection marker flanked by 2.2 kb of 5′UTR and

0.55 kb of 3′UTR of the pb*dhfr*-ts were introduced into the *CSP* and *TRAP* loci. Upon successful integration of the CSΔN construct, the wild-type copy of *CSP* is replaced by a truncated version lacking the amino acids NKSIQAQRNLNELCYNEGNDNKLYHVLNSKNGKIYNRNTVNRLLADAPEGK KNEK KNEKIERNNKLKQP, which encompass the entire N-terminal region, except for the signal sequence. The successful integration of the TRAP-VAL construct changes the canonical rhomboid protease cleavage site motif AGGIIGG to VALIIGV. The transfected parasite populations were cloned. *CSP* and *TRAP* loci of wild-type control, CSΔN and TRAP-VAL parasite clones were genotyped by PCR to verify successful recombination and the presence of the desired mutations. Integration of the *CSΔN* copy at the correct location was verified by PCR of PbmCherry CSΔN clone B3 parasite line using previously described primer sets ([*Coppi et al., 2011*]; see *Supplementary file 1* for primer sequences). PbmCherry TRAP-VAL clone A3 was genotyped by PCR using previously described primer sets ([*Ejigiri et al., 2012*]; see *Supplementary file 1* for primer sequences). For both mutants, the presence of the desired mutation was confirmed by sequencing.

## Western blotting

Wild-type control and CSΔN salivary gland sporozoites were lysed in reducing sample buffer, and $1.5 \times 10^4$ sporozoites per lane were loaded and separated by SDS-PAGE and transferred to nitrocellulose membrane. The membrane was cut in between the markers for 52- and 76-kDa, and the top half was incubated with a 1:100 dilution of rabbit polyclonal antisera to the repeat region of TRAP, used as a loading control (*Ejigiri et al., 2012*), followed by anti-rabbit IgG conjugated to horseradish peroxidase (HRP). The bottom half of the blot was incubated with 1 µg/ml mouse monoclonal CSP antibody (clone 3D11, [*Yoshida et al., 1980*]), followed by anti-mouse IgG, conjugated to HRP. HRP was visualized using enhanced chemiluminescence (GE Healthcare, United Kingdom).

## Immunofluorescence

Wild-type control and CSΔN salivary gland sporozoites were fixed with 4% paraformaldehyde, blocked in 3% BSA and detected with a 1:100 dilution of polyclonal rabbit N-terminal CSP serum (*Coppi et al., 2011*) or with 1 µg/ml mouse monoclonal CSP antibody (clone 3D11, [*Yoshida et al., 1980*]), followed by detection with Alexa-Fluor-488-conjugated anti–rabbit or anti-mouse IgG (Life Technologies, Durham, NC, United States). Samples were preserved in Prolong Gold mounting medium (Life Technologies) and imaged using phase-contrast and fluorescence microscopy on an upright Nikon E600, and images were acquired with a digital camera (DsRi1 Nikon).

## Determination of parasite liver load

4- to 6-week-old Swiss Webster mice were infected with $10^4$ wild-type control or CSΔN salivary gland sporozoites, either by intravenous or intradermal injection. For intradermal inoculation, mice were lightly anaesthetized with ketamine/xylazine and maintained at 37°C on a slide warmer. Sporozoites were injected into the ear pinna in a total volume of 0.2 µl DMEM with a Flexifill microsyringe (World Precision Instruments, Sarasota, FL, United States). 36.5 hr after sporozoite inoculation, the livers of all mice were harvested, total RNA was isolated using Tri-reagent (Molecular Research Center, Cincinnati, OH, United States) and the parasite burden in the liver was quantified using reverse transcription (RT) with random hexamers, followed by real-time PCR with the SYBR green Kit (Applied Biosystems, Carlsbad, CA, United States) using primers specific for *P. berghei* 18S rRNA as outlined previously (*Bruña-Romero et al., 2001*). 10-fold dilutions of a plasmid construct containing the *P. berghei* 18S rRNA gene were used to create a standard curve.

## Determination of prepatent period

Swiss Webster mice were lightly anesthetized by intraperitoneal injection of ketamine/xylazine and exposed for 30 min to the bites of *A. stephensi* infected with wild-type control or TRAP-VAL parasites. The onset of blood stage infection was determined by observation of Giemsa-stained blood smears, beginning on day 3 after inoculation.

## Intravital imaging

Sporozoites in the ear pinna of mice were imaged similar to a previously described protocol (*Amino et al., 2007*). 4- to 5-week-old C57BL/6 mice from Taconic were lightly anesthetized by intraperitoneal injection of ketamine/xylazine (35–100 µg ketamine/gram body weight), and sporozoites were injected intradermally into the dorsal sheet of the ear pinna in a total volume of 0.2 µl using a NanoFil-10 µl

syringe with a NF33BV-2 needle (World Precision Instruments). All experiments were completed within 60 min of salivary gland dissection to ensure that imaging and parasite conditions remained constant. By taping the ear pinna to a coverslip, the mouse was mounted on the platform of an inverted Zeiss Axio Observer Z1 microscope with a Yokogawa CSU22 spinning disk and a preheated temperature-controlled chamber at 30°C. For imaging of mosquito-bite inoculated sporozoites, infected mosquitoes were kept in a plexiglass tube with mesh screen and were allowed to probe in the portion of the ear pinnae immobilized on the microscope stage. Parasites were visualized using a 10× objective and magnified with a 1.6 Optovar, resulting in an x and y imaging volume of 500 μm × 500 μm. Z-slices were imaged with an exposure time of approximately 150 ms and time-lapse stacks with 3–5 slices spanning a total depth of 30–50 μm were captured at approximately 1 Hz over a total time of 4 min using an EMCCD camera (Photometrics, Tucson, AZ, United States) and 3i slidebook 5.0 software. The indicated time points after inoculation refer to the starting time of the 4-min acquisition. Images were projected into a single z-layer and the resulting two-dimensional data set was processed and manually tracked using Imaris software (Bitplane, Concord, MA, United States). For visualization of dermal blood vessels, mice were intravenously injected with 15 μg of rat anti-mouse CD31 coupled to Alexa-Fluor-647 (clone 390, Biolegend, San Diego, CA, United States), 2–4 hr prior to imaging.

## Manual tracking and data analysis
### Statistical analysis
Statistical analysis of scatter dot plots were analyzed using unpaired Student's $t$-test and statistical significance between samples is indicated with asterisks as follows *p < 0.05; **p < 0.01; ***p < 0.001; ****p < 0.0001; ns, nonsignificant (p > 0.05). Track curvature analysis was tested for statistical significance using Kolmogorov–Smirnov two-sample test.

### Track projections
Reconstructed sporozoite tracks were plotted to a common origin using the Imaris MATLAB Xtension ('plot tracks to common origin').

### MSD
The MSD was calculated for parasites that exhibited motility and tracks spanning less than the total video time of 4 min were discarded. The MSD of wild-type control sporozoites over time included the following data points: 5 min (2 videos, 37 tracks), 10 min (14 videos, 179 tracks), 20 min (7 videos, 95 tracks), 30 min (9 videos, 129 tracks), 60 min (8 videos, 77 tracks) and 120 min (7 videos, 67 tracks). In order to be able to analyze the MSD from sporozoite populations obtained from videos with varying frame rates, the data were interpolated to a common time interval using the cubic spline function in GraphPad prism. Linear regression fitting was performed using GraphPad Prism software and suggested a linear relationship between MSD and time. To obtain the MSD of CSΔN and TRAP-VAL sporozoites, 179 wild-type control sporozoites were tracked in 12 videos and compared to 211 tracks obtained from 29 videos of CSΔN parasites and 215 tracks of TRAP-VAL sporozoites originating from 33 videos; all videos for this analysis were acquired 10 min after intradermal inoculation of sporozoites.

To generate the MSD of sporozoites moving in proximity to CD31-labeled vessels 10 min after intradermal injection, 18, 25, or 24 videos of wild-type control, CSΔN and TRAP-VAL sporozoites, respectively, were analyzed, and 313, 352, or 192 tracks of sporozoites moving in proximity to CD31-labeled vessels were generated. 163, 121, or 89 tracks of sporozoites moving in avascular areas of the dermis were produced from the same set of videos of wild-type control, CSΔN and TRAP-VAL sporozoites, respectively. To calculate the MSD of the obtained tracks with various duration times, all tracks were uniformly fragmented into tracks of 30 s in duration.

### Probability distribution
The x and y position data of sporozoite tracks were exported, and the total displacement exhibited by sporozoites at the end of the 4-min videos, defined as the Euclidean distance between first and last track location, was calculated. The probability distribution P(r) of the final displacements was obtained by binning displacement data and dividing each bin by the total number of tracks to create a frequency-normalized histogram. Data analysis was performed via custom-written MATLAB software (https://github.com/kevinkchiou/Malaria-Motility).

## Apparent speed

Apparent speed has previously been discussed (*Kan et al., 2014*) as an estimation of the instantaneous speed and is calculated as the sum of distances between track locations, divided by the track duration. This measure is highly sensitive to the frame rate with which the video was acquired (*Kan et al., 2014*) and to achieve consistent interval times across the entire data set, for the calculation of apparent speed we utilized only positional data in 3-s intervals, which is larger than the longest interval time present in the data set.

## Track straightness

The track straightness is defined as the ratio of displacement to total track length and as a result can range between zero (entirely constrained track) and one (the track is a straight line) (*Beltman et al., 2009*). Notably, this ratio is highly sensitive to the track duration (*Beltman et al., 2009*) and therefore only tracks with the same duration were compared. To calculate the track straightness of wild-type control sporozoites over time, the tracks of meandering and linearly moving sporozoites tracked for the MSD analysis were used. To account for the percentage of circling sporozoites at the individual time point, 80 sporozoites continuously moving in the same circle were tracked, and the average track straightness of exclusively circling sporozoites was found to be 0.03. A number of tracks with this value were added to the analysis, representative for the percentages of circling sporozoites observed at the individual time point.

## Calculation of time spent in proximity to CD31-labeled vessel

To obtain duration of motility near blood vessels in videos acquired 10 min after intradermal inoculation, motile sporozoites were tracked when observed to be circling around or gliding alongside CD31-labeled vessels. In a given video, the durations of all tracks were added and expressed as percentage of total motile time, which was calculated as number of motile sporozoites multiplied by the total video time. This analysis was performed on 33 videos of wild-type control, 31 videos of CSΔN, and 35 videos of TRAP-VAL sporozoites.

## Track curvature analysis

The tracking data originating from manual differential tracking of sporozoites moving in proximity to CD31-labeled vessels as well as in avascular areas of the dermis were used to determine the curvature of these two sets of sporozoite tracks. To find the local curvature $\kappa_i$ at data point $i$ of a two-dimensional track, a circle is fit to position data along that track that are within a finite time window $T = 10$ s. In this case, the track curvature is the inverse of the fit radius, $\kappa_i = r_i^{-1}$. The time window $T$ serves two purposes: (1) to average out positional noise and (2) to maintain consistent time windows between data sets with different time intervals.

Consider a data point indexed by $i$. We denote the time–space coordinates of this point as $(t_i, x_i, y_i)$. We find all points along this track within $t_i \pm T/2$ of which there are $N$. We denote the resulting coordinates of all those points as $(t_1, x_1, y_1), \ldots, (t_N, x_N, y_N)$. The time points associated with these positions all lie within $t_i \pm T/2$ and satisfy the relation $t_i - T/2 \leq t_1 < \ldots < t_N \leq t_i + T/2$. For a given data point, if this window exceeds the available track data, then we omit it and perform no fitting. If the track data properly spans the time window around $t_i$, then we collect the $N \geq 3$ points and fit a center and radius $(x_c, y_c, r)_i$ to the positions. The fit is performed via linear least-squares regression as detailed below. The general equation of a circle in the $(x, y)$ plane centered at $(x_c, y_c)$ with radius $r$ is

$$(x - x_c)^2 + (y - y_c)^2 = r^2.$$

The least-squares regression for a set of positional data $\{(x_i, y_i)\}$ of size $N$ to a circle benefits from the fact that the distance of the closest approach is along the radial direction. As a result, the least-squares residual is

$$R^2 = \sum_{i=1}^{N} \left( \sqrt{(x_i - x_c)^2 + (y_i - y_c)^2} - r \right)^2.$$

This equation is non-quadratic in the circle parameters $(x_c, y_c, r)$; therefore no linear regression solution exists as written (although non-linear regression techniques can be applied).

However, a reorganization of the equation for a circle and computing the squared residual of $r^2$ (instead of $r$) allows for solution by linear regression. Expanding as

$$x^2 + y^2 = r^2 - x_c^2 - y_c^2 + 2xx_c + 2yy_c,$$

we designate a new parameter $\alpha = r^2 - x_c^2 - y_c^2$. The modified residual is then

$$R_{\mathrm{mod}}^2 = \sum_{i=1}^{N} \left( \alpha + 2x_i x_c + 2y_i y_c - x_i^2 - y_i^2 \right)^2.$$

Within the sum of squares, the terms are all either (1) linear with respect to the parameters $(x_c, y_c, \alpha)$ or (2) constant. This is solvable by linear regression on the circle parameters.

We perform said linear regression by solving the associated matrix equation. Let us define the vectorized parameter $z = (x_c, y_c, \alpha)$. Residual minimization over $z$ becomes

$$\min_z \ R_{\mathrm{mod}}^2 = \min_z \| A_{ij} z_j - b_i \|^2$$

where $j \in \{1, 2, 3\}$ indexes the fit parameters $(x_c, y_c, \alpha)$, and $i \in \{1, \ldots, N\}$ indexes the data entries. Then best-fit parameters $\bar{z}$ that minimize this residual are then $\bar{z} = (A^T A)^{-1} A^T b$ where,

$$\bar{z} = \begin{pmatrix} \bar{x}_c \\ \bar{y}_c \\ \bar{\alpha} \end{pmatrix}, A = \begin{pmatrix} 2x_1 & 2y_1 & 1 \\ \vdots & \vdots & \vdots \\ 2x_N & 2y_N & 1 \end{pmatrix}, b = \begin{pmatrix} x_1^2 + y_1^2 \\ \vdots \\ x_N^2 + y_N^2 \end{pmatrix}.$$

Here, $A^T$ denotes the transpose of the matrix $A$, and $(A^T A)^{-1}$ is the matrix inverse of $A^T A$.

Once best-fit values $(\bar{x}_c, \bar{y}_c, \bar{\alpha})$ are obtained, the best fit curvature $\bar{\kappa}$ can be computed by its relation to $\bar{z}$:

$$\bar{\kappa} = \sqrt{ \left( \bar{\alpha} + \bar{x}_c^2 + \bar{y}_c^2 \right)^{-1} }.$$

Data analysis was performed via custom-written MATLAB software, available on GitHub at https://github.com/kevinkchiou/Malaria-Motility. The collection $\{\bar{\kappa}_i\}$ computed by this algorithm are aggregated and plotted as a histogram.

## Acknowledgements

We thank the Johns Hopkins Malaria Research Institute parasite core and insectary, especially Christopher Kizito and Dr Godfree Mlambo for providing *P. berghei*-infected mosquitoes. We are grateful to Johns Hopkins University School of Medicine Microscope Facility for their assistance with imaging experiments, to Dr Ian Cockburn for valuable input into the intravital imaging technique and to Diego Espinosa and Lirong Shi for technical assistance. We also thank Amanda Balaban for critical reading of the manuscript and Drs Fidel Zavala and Friedrich Frischknecht for helpful discussions. This work was supported by the National Institutes of Health grant A1056840 (PS), by a Johns Hopkins Malaria Research Institute fellowship (CSH) and NSF grant DMR-1104637 (KC).

## Additional information

### Funding

| Funder | Grant reference | Author |
| --- | --- | --- |
| National Institutes of Health (NIH) | A1056840 | Photini Sinnis |
| Johns Hopkins Malaria Research Institute | postdoctoral fellowship | Christine S Hopp |
| National Science Foundation (NSF) | DMR-1104637 | Kevin Chiou |

The funders had no role in study design, data collection and interpretation, or the decision to submit the work for publication.

## Author contributions

CSH, Conception and design, Acquisition of data, Analysis and interpretation of data, Drafting or revising the article; KC, PS, Conception and design, Analysis and interpretation of data, Drafting or revising the article; DRTR, AJL, Analysis and interpretation of data, Drafting or revising the article; AMS, SMK, Drafting or revising the article, Contributed unpublished essential data or reagents

## Ethics

Animal experimentation: All animal work was conducted in accordance with the recommendations by the Johns Hopkins University Animal Care and Use Committee (IACUC), under the IACUC-approved protocol #M011H467.

## Additional files

### Supplementary file

• Supplementary file 1. Primer sequences used to generate and confirm the transgenic and mutant parasites described in this study.

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
