## [Decision Letter]

Thank you for sending your work entitled “Longitudinal analysis of *Plasmodium* sporozoite motility in the dermis reveals component of blood vessel recognition” for consideration at *eLife*. Your article has been favorably evaluated by Prabhat Jha (Senior Editor), a Reviewing Editor, and three reviewers.

Your study regarding migration of dermally delivered *Plasmodium* sporozoites received favorable reviews from all the reviewers. The results convincingly show the difference between “avascular” and “vascular” migration of the parasites using both wild type and mutant parasites. The data acquisition is deemed accurate and the quantified data carefully evaluated. Overall, as a technical tour-de-force, the study provides important new insights into in vivo malaria sporozoite behavior during the critical interval between injection into the dermis and entry into blood vessels. The reviewers stated that the quality of the experiments and the importance of the observations merit an acceptance of the manuscript.

Notwithstanding, the reviewers had several major concerns, and collectively it was decided that the final acceptance is pending your responses to their comments.

1) Several questions arose in regard to the use of CD31 labeling. A concern was expressed whether labeling with CD31 might have interfered with sporozoite migration and invasion of the vasculature. Another concern was specificity. Because CD31 also labels lymphatic vasculature, the authors need to document sporozoite invasion into each of the CD31+ vessels. Otherwise, these cannot be called blood vessels or the motility cannot be referred to as vascular motility. This should be corrected throughout the text. Finally, there was concern considering the method of analysis. The proximity of the parasites to the CD31+ vessels was scored manually, which can lead to errors. Therefore, it would be helpful if the authors would perform analyses on one of the 3D datasets to validate that the numbers obtained from the maximum intensity projections are correct.

2) The interpretation that *Plasmodium* parasites switch the mode of their motility to engage blood vessel should be toned down. This observation has previously been reported and the current data proposing such a change from fast to slow when engaging vessels are not convincing. For the quantification of the parasites that are changing the mode of motility it is essential to consider that while some parasites are losing straight/fast mode, the numbers of the parasites with curved/slow mode of motility is not increasing. Hence, it is the proportion, rather than the number, that changes.

3) It appears that some of the work has not been cited properly and some previously reported observations have been excluded from the Discussion. Among others, the authors should include published works on sporozoite gliding by [25], work by [1] on decreased sporozoite speed on contacting blood vessels and work by Battista, Frischenechet and Schwarz (2014) on the curvature of sporozoites fitting the capillaries. The current results should be discussed further in the context of observations reported by Hellman (2011). It is suggested that the authors consider the possibility that the current data could be explained by simple physical constraints analogous to the microfabricated pillers in the Hellman paper.

4) Another comment relates to the mutant parasites that were shown to be impaired in their ability to invade blood vessels, despite the fact that they found blood vessels and remained associated with them. Although the authors do discuss it, it appears that the discussion may be inadequate in providing a cogent explanation for this observation.

5) A revision of the title has been suggested so that it precisely reflects the findings. A clarification should be provided – in your already well-written manuscript – about the use of the term “vascular,” which does mean that the sporozoites are inside the vessels.

6) A rather important question was posed regarding different migration patterns caused by the sporozoite delivery route: by a mosquito bites versus a syringe.

7) The legends need to include more detail about the identification of the sporozoites that invade the vessels and the video should be labeled more precisely.

---

## [Author Response]

We have responded to the editor’s request to move all supplemental methods into the main body of the paper. This applied to the methodology for generating and verifying the mutants and to the mathematical analysis used for track curvature analysis, which are now in the Methods section of the manuscript.

1) Several questions arose in regard to the use of CD31 labeling. A concern was expressed whether labeling with CD31 might have interfered with sporozoite migration and invasion of the vasculature.

We agree with reviewers that interference of CD31-labeling with sporozoite motility and exit from the dermis is a valid concern. In the original version of the manuscript, in Figure 5–figure supplement 2, we showed that CD31-labeling did not interfere with entry into dermal blood or lymphatic vessels, as the sporozoite invasion rates are similar in videos acquired in mice with and without CD31-labeled blood vessels. However, we had not looked at the effect of CD31 labeling on sporozoite migration. In part, this was because vessels were not labeled in the majority of the videos used to determine MSD, they were only labeled in a subset of the 10-minute videos, since this was the time point used for quantification of vessel entry. We went back to the videos used for the MSD analysis (Figure 1) and reanalyzed the 10 minute time point data to separately calculate the MSD of sporozoites in videos with and without CD31-labeling. We found that the MSD in the two populations was similar and these data are now shown as a new panel of Figure 5—figure supplement 1.

Another concern was specificity. Because CD31 also labels lymphatic vasculature, the authors need to document sporozoite invasion into each of the CD31+ vessels. Otherwise, these cannot be called blood vessels or the motility cannot be referred to as vascular motility. This should be corrected throughout the text.

The reviewers are correct that CD31 is a pan-endothelial marker, expressed by both lymphatic and blood vessel endothelial cells, although the expression level on lymphatic vessels is lower than on blood vessels (Podgrabinska et al., Molecular characterization of lymphatic endothelial cells, PNAS, Vol 99:16069, 2002). It is important to note, and we neglected to do so in the original manuscript, that the methodology we used does not result in CD31 staining of lymphatic vessels. We injected the anti-CD31 intravenously into the blood circulation 2-3 hours prior to the imaging experiment, and as a result it primarily stained blood vessels, since at this point the antibody has not crossed into the extravascular space to be drained into the lymphatic system. For imaging of lymphatic vessels, labeling antibodies are typically injected a day prior to the imaging experiment (Sarkisyan et al., Real-time differential labeling of blood, interstitium, and lymphatic and single-field analysis of vasculature dynamics in vivo, Am. J. Physiol. Cell Physiol., 2012). Included as a new figure supplement, Figure 2—figure supplement 1, is a still image obtained from live microscopy of a mouse ear in which anti-LYVE-1 antibody, specific for lymphatics, was inoculated intravenously 24 hours prior to imaging and anti-mouse CD31 antibody was inoculated 3 hours prior to imaging. No co-localization is observed, indicating that in our protocol, antibodies to CD31 do not label lymphatics. Lastly, given the unique structural characteristics of lymphatic capillaries, which possess frequent blind endings and are wider and flatter than blood vessels (Tripp et al., The lymph vessel network in mouse skin visualized with antibodies against the hyaluronan receptor LYVE-1, Immunobiology, 213:715–28, 2008), we are confident that the vessels stained in our experiments are blood and not lymphatic vessels. We have changed the manuscript to clarify this important issue (see the subsection “Sporozoites recognize dermal CD31+ blood vessels”).

Finally, there was concern considering the method of analysis. The proximity of the parasites to the CD31+ vessels was scored manually, which can lead to errors. Therefore, it would be helpful if the authors would perform analyses on one of the 3D datasets to validate that the numbers obtained from the maximum intensity projections are correct.

We agree with the reviewers that 3D tracking would be ideal and that tracking sporozoites in the maximum projections may lead to errors. However, the acquired stacks have virtually no 3D resolution. Given the high gliding speed of sporozoites, entire stacks have to be acquired with short interval times, and since exposure times of up to 200 ms per capture are necessary to visualize the parasites in the dermis, we were only able to acquire 3-5 slices using a 10x objective with maximal pinhole setting. This amounted to only semi-confocal data, with essentially no z-resolution, which does not allow for tracking in three dimensions. While we agree that in individual cases we may have scored sporozoites as being close to vessels when they were not, given the number of data points we acquired, we do not think that this is a problem, particularly since these false positives would only minimize the differences between vascular and avascular motility, as this would not be an issue with sporozoites moving far from vessels.

*2) The interpretation that* Plasmodium *parasites switch the mode of their motility to engage blood vessel should be toned down. This observation has previously been reported and the current data proposing such a change from fast to slow when engaging vessels are not convincing.*

The reviewers are correct, the decrease in sporozoite gliding speed upon approach of a blood vessel has previously been documented (Amino et al., Nat. Med., 2006) and we have now discussed this previous finding in the Discussion: “These data build upon and solidify the previous observation of Amino et al., who quantified the speed of a small number of sporozoites gliding around blood vessels and reported a decrease after contact with the vessel”. We agree that the initial seminal study of Amino et al. is the first observation of this slow-down, however this study only looked at a few sporozoites, while our results are based on hundreds of measurements, thus really solidifying this finding. We were surprised to read the statement of the data not being convincing, since the editor’s letter stated: “The results convincingly show the difference between “avascular” and “vascular” migration […] the study provides important new insights into in vivo malaria sporozoite behavior during the critical interval between injection into the dermis and entry into blood vessels.” While the change in speed upon approach of a blood vessel is moderate, it is highly significant (Figure 2) and hopefully the reviewers are persuaded to agree.

For the quantification of the parasites that are changing the mode of motility it is essential to consider that while some parasites are losing straight/fast mode, the numbers of the parasites with curved/slow mode of motility is not increasing. Hence, it is the proportion, rather than the number, that changes.

We completely agree with the reviewers and now include this information. We expressed the data as percentages because many of our videos were not part of complete time courses. Thus, in many instances we do not know the proportion of sporozoites that remain in the field at the late time points compared to earlier time points. Nonetheless, we do have two complete time courses, in which the same site was imaged from 5 to 120 minutes. These data were included in the original analyses shown in Figure 1. We now use them to address the reviewers’ concern and have generated a supplement to Figure 1 (Figure 1—figure supplement 1). Two things become evident when we visualize the data this way. First, at 20 and 30 minutes after inoculation, while the total number of sporozoites and the number of non-motile sporozoites does not change significantly, the number of sporozoites moving in a linear fashion decreases and the number that are circling continuously throughout the video increases. These data are consistent with our observation of an initial period of high dispersal motility, followed by more constrained motility. However, when we look at the very late time points, 60 and 120 minutes after inoculation, these observations are confounded by the 20% and 40% fewer sporozoites in the field respectively, and by the significantly increased number of non-motile sporozoites. We have therefore added more discussion of these data in the Results section and in the Discussion: “In contrast to previous reports of apparently random motility […] indicates that there is a specific recognition event that leads to this change.”

*3) It appears that some of the work has not been cited properly and some previously reported observations have been excluded from the Discussion. Among others, the authors should include published works on sporozoite gliding by*
[25]*, work by*
[1]
*on decreased sporozoite speed on contacting blood vessels and work by Battista, Frischenechet and Schwarz (2014) on the curvature of sporozoites fitting the capillaries.*

We have inserted references to the listed studies and more fully discussed the studies we had already cited. The Vanderberg et al. study (Int. J. Parasitol., 2004) is cited with other reports of sporozoite motility being important for dermal exit. We now mention the previous report of decreased sporozoite gliding speed in proximity to blood vessels (Amino et al., Nat. Med., 2006) in the Discussion. Additionally, we added a discussion on early descriptions of sporozoite geometry and how an in silico model, predicting the sporozoite's ability to change its curvature (Battista et al., Phys. Rev. E. Stat. Nonlin. Soft. Matter Phys., 2014) is supportive of our data.

The current results should be discussed further in the context of observations reported by Hellman (2011). It is suggested that the authors consider the possibility that the current data could be explained by simple physical constraints analogous to the microfabricated pillers in the Hellman paper.

The reviewers make a valid point that it has previously been discussed that the distance between obstacles in vitro and different dermal environments in vivo can induce changes in sporozoite motility, which leads to the hypothesis that obstacle distances in the dermis guide sporozoite migration (Hellman et al., PLoS Pathog., 2011). However, we are reporting the first evidence of the ability of the sporozoite to alter motility in response to a specific host structure, suggesting sporozoite migration is responsive to host signals more complex than simple physical constraints. We think that more intravital imaging needs to be performed with visualization of components of the extracellular matrix and host cells other than endothelial cells in order to shed light on the host-pathogen interactions relevant for blood vessel recognition. However, we have extended the Discussion to include the findings of Hellman et al., and to allow the reader to put our findings into the perspective of what has previously been found.

4) Another comment relates to the mutant parasites that were shown to be impaired in their ability to invade blood vessels, despite the fact that they found blood vessels and remained associated with them. Although the authors do discuss it, it appears that the discussion may be inadequate in providing a cogent explanation for this observation.

We agree that there was not a cogent explanation for this and, in large part, it is because more work is needed to fully explain this observation. We have, however, modified and added to our original text to improve clarity and flesh out some of our hypotheses as to what could cause the phenotype that we observe. Please see in the Results section: “These data point to an additional impairment in blood vessel entry, i.e. even when they do contact blood […] results in their inability to complete the entry process.”

5) A revision of the title has been suggested so that it precisely reflects the findings. A clarification should be provided – in your already well-written manuscript – about the use of the term “vascular,” which does mean that the sporozoites are inside the vessels.

The question of the title goes back to point number 4. We believe that the current title accurately reflects our findings. Regarding the term “vascular” – we now see the confusion that this could cause. We have gone back and changed all such uses of the word to “perivascular” which should clarify its meaning.

6) A rather important question was posed regarding different migration patterns caused by the sporozoite delivery route: by a mosquito bites versus a syringe.

We agree that this is an important point and had included a preliminary analysis of sporozoite migration patterns after inoculation by mosquito bite in the original manuscript (Figure 1—figure supplement 2). For this figure, we combined two imaging time courses of mosquito-inoculated sporozoites and analyzed their MSD and motility patterns. As shown, we found no significant difference in sporozoite behavior in terms of MSD and proportion of manually counted sporozoite motility patterns. We have also added more commentary on these data, at the end of the first section of the Results.

7) The legends need to include more detail about the identification of the sporozoites that invade the vessels and the video should be labeled more precisely.

We have amended the legend to Figure 5, as well as the fifth section of the Results, to give more details on the identification of invasion events. Compared to previous video material (Amino et al., Nat. Med., 2006), blood vessel invasion events in videos that include CD31-labeling are not always characterized by the sudden increase in speed, which is a result of our slower acquisition speed (due to the capture of two channels). The sporozoite in circulation is frequently carried out of the field of view before the next frame is captured. We therefore altered our description of the manual invasion counts and now define blood vessel invasion by “a sudden increase in speed or visual entry of the blood vessel and disappearance out of the field of view”. Since the frame rate with which images were acquired was higher in videos without CD31-labeling, these videos allow the viewer to see the sudden increase in speed. We therefore included an additional video (Video 8), in which two blood vessel invasion events, characterized by the sudden increase in speed, are clearly visible.